# AND/OR Branch-and-Bound for
# Computational Protein Design Optimizing K*

**Bobak Pezeshki**[1]    **Radu Marinescu**[2]    **Alexander Ihler**[1]    **Rina Dechter**[1]

[1]University of California, Irvine
[2]IBM Research

## Abstract

The importance of designing proteins, such as high affinity antibodies, has become ever more apparent. Computational Protein Design can cast such design problems as optimization tasks with the objective of maximizing K*, an approximation of binding affinity. Here we lay out a graphical model framework for K* optimization that enables use of compact AND/OR search algorithms. We designed an AND/OR branch-and-bound algorithm, AOBB-K*, for optimizing K* that is guided by a new K* heuristic and can incorporate specialized performance improvements with theoretical guarantees. As AOBB-K* is inspired by algorithms from the well studied task of Marginal MAP, this work provides a foundation for harnessing advancements in state-of-the-art mixed inference schemes and adapting them to protein design.

## 1  INTRODUCTION

The powerful framework of graphical models allows for reasoning about conditional dependencies over many variables. The Marginal MAP (MMAP) task finds an optimal configuration of a subset of variables (MAP variables), that have the highest marginal probability. We define a new related task, K*MAP, which instead asks for the configuration of MAP variables that maximizes a quotient of marginals of conditionally disjoint subsets of the remaining variables. This ratio is known as K* [Lilien et al., 2004]. In computational protein design (**CPD**), K* estimates affinity between interacting subunits. Maximizing K* corresponds to maximizing the likelihood that the subunits will associate [Hallen and Donald, 2019].

Like MMAP, K*MAP distinguishes between maximization (MAP) variables and summation (SUM) variables. Moreover, the SUM variables are further partitioned into a subset whose marginal corresponds to the numerator of the K* ratio and a subset corresponding to the denominator. Like MMAP, the K*MAP problem relies on max-sum inference and more difficult than either pure max- or sum- inference tasks as the sum and max operations do not commute. This forces constrained variable orderings that may have significantly higher induced widths [Dechter, 1999, 2019]. This also implies larger search spaces when using search algorithms or larger messages when using message-passing schemes. Furthermore, bounding K* requires both upper and lower bounding of marginals, producing an additional challenge.

Over the last several years, there have been several advances in algorithms for solving MMAP [Lou et al., 2018a,b, Marinescu et al., 2019, 2018a,b], many of which can potentially be adapted for K*MAP. Our objective in this work is to (1) create a framework for which future advancements to the K*MAP task can be made leveraging these powerful algorithms, (2) to begin an exploration of new bounded heuristics for the task, and (3) to create a foundation for efficient algorithms solving (and eventually approximating) the K*MAP. As such, this work presents the following contributions:

1. Two formulations of K*MAP as a graphical model

2. wMBE-K*, a weighted mini-bucket algorithm for K*MAP enhanced with a domain partitioning scheme

3. AOBB-K*, a depth-first branch-and-bound algorithm over AND/OR search spaces for solving K*MAP

4. A thresholding scheme to exploit determinism accompanied with correctness guarantees

5. Extensive empirical analysis comparing these schemes to state-of-the-art BBK* illustrating their potential

## 2  BACKGROUND

Here we provide a brief background about protein design and graphical models.

*Accepted for the 38th Conference on Uncertainty in Artificial Intelligence* (UAI 2022).

## 2.1 COMPUTATIONAL PROTEIN DESIGN

Proteins are complex macromolecules composed of chains of amino acids that fold into a three dimensional structure based on intra- and inter- molecular interactions. Proteins are essential to life and act as the machinery for a myriad of biological functions including chemical metabolism, cellular regeneration, immune response, movement, and structural support. Computational Protein Design (**CPD**) is a field focused on designing new proteins (ie. new amino acid sequences) to satisfy desired objectives [Gainza et al., 2016]. Two sub fields of CPD are *de novo* protein design: the generation of novel protein sequences unrelated to those found naturally [Huang et al., 2016], and protein redesign: the modification of known proteins to alter their function or interactions [Gainza et al., 2016]. The task we focus on in this work is this latter sub field of protein redesign, specifically for improving protein subunit interactions. For simplicity, we will use CPD to refer to such redesign tasks.

A somewhat orthogonal task often confused with protein design is protein structure prediction, or PSP, which aims to predict the three-dimensional structure resulting from a *given* amino acid sequence. PSP gained visibility recently with the advancements of AlphaFold [Jumper et al., 2021] and can be leveraged by certain protein design tasks.

For the protein redesign tasks that this work focusses on, certain amino acid positions (or **residues**) of a protein-of-interest are deemed as mutable - these are amino acid positions where different amino acid mutations will be considered - and through a computational process, a preferred sequence is determined [Donald, 2011]. Throughout the process, sets of mutations are explored, each comprising a specific amino acid sequence. Given a sequence (or in some methods, even partial sequence) an estimate of the resulting protein's goodness can be estimated. This goodness is determined by considering the possible conformations of the protein's backbone and amino acid side-chains. The state space for these conformations is continuous (and even when discretized, is extremely large) leading to an intractable problem.

To remedy, simplifications can be made: (i) consider only a subset of residues as mutable, (ii) discretize side-chain conformations into rotamers, and (iii) assume a fixed protein backbone conformation. With these simplifying assumptions, many algorithms have been designed to find mutations that can potentially result in improved protein functionality [Hallen and Donald, 2019, Zhou et al., 2016].

## 2.2 K* AND K*MAP

The affinity between two protein subunits $P$ and $L$ relates to the rate at which they bind into a complexed form $PL$ and dissociate reforming $P$ and $L$ (as indicated by the chemical equation: $P + L \rightleftharpoons PL$). This said equilibrium is associated with a constant, $K_a$, and can be determined in vitro by computing the ratio of persisting concentrations of each species by $K_a = \frac{[PL]}{[P][L]}$ [Rossotti and Rossotti, 1961]. However, in order to compare $K_a$ values of various designs in vitro, it is necessary to synthesize the protein subunits through molecular processes that are both timely and costly.

$K_a$ can also be approximated as $K^\int = \frac{Z_{PL}^\int}{Z_P^\int Z_L^\int}$, where $Z_\gamma^\int = \int_{\mathcal{C}} e^{-\frac{E_\gamma(c)}{\mathscr{R}T}} dc$ and $Z_{PL}^\int$, $Z_P^\int$, and $Z_L^\int$ are approximations of partition functions of the bound and unbound form of the protein - namely, when the protein subunits are complexed (denoted $PL$) and when they are dissociate (denoted $P$ and $L$), respectively - that capture the entropic contributions of their various conformations $\mathcal{C}$ [Hill, 1987, McQuarrie, 2000]. $E_\gamma(c)$ represents the energy of a particular conformation $c$ of the protein in form $\gamma \in \varphi$ where $\varphi = \{P, L, PL\}$, $\mathscr{R}$ is the universal gas constant, and $T$ is temperature (in Kelvin). We can further use a model that discretizes the conformation space, denoted as $D(C)$. This computed estimate is known as K* [Lilien et al., 2004]:

$$K^* = \frac{Z_{PL}}{Z_P Z_L}, \quad Z_\gamma = \sum_{c \in D(C)} e^{-\frac{E_\gamma(c)}{\mathscr{R}T}} \quad (1)$$

Due to independence of residue interactions between the different forms $\gamma$ of the protein, we can generalize further as $K^* = \frac{Z_B}{Z_U}$, where $B$ represents the bound (complexed) form(s) and $U$ represents the unbound (dissociate) forms. (For a two-subunit system, $B = \{PL\}$ and $U = \{P\} \cup \{L\}$). This generalized representation can be used for K* computations involving more than two subunits.

A common goal in protein redesign is to maximize protein-ligand interaction. Previously, this was done by minimizing an objective called the GMEC (global minimum energy conformation) over only the complexed protein form $PL$ [Ruffini et al., 2021, Hallen and Donald, 2019, Zhou et al., 2016]. The GMEC, a pure minimum over energies of the complex's conformations, ignores the realization that protein structures are dynamic. Also, by the GMEC focusing only on the complexed form, it ignores dynamicity of subunit associations. However, since minimizing the GMEC results in a pure optimization task - much easier than that of mixed inference - many solvers use this objective. Alternatively, the stronger K* objective (contrasted in Lilien et al. [2004]) captures both the dynamicity of protein conformations and subunit interactions. K*MAP is the formalization of computational protein design as a task to maximize K*,

$$K^*\text{MAP} = \underset{\boldsymbol{R}}{\operatorname{argmax}} K^*(r) \quad (2)$$

looking for amino acid assignments $\boldsymbol{R} = r$ that maximize K*. The goal of recent work, and that presented here, is to develop efficient algorithms for computing K*MAP to predict a small set of promising sequences to experiment on in vitro and in vivo, saving great time and cost.

## 2.3 GRAPHICAL MODELS AND THEIR AND/OR SEARCH SPACES

Our work taps into algorithms developed for the MMAP task defined on graphical models. A **graphical model**, such as a Bayesian or Markov network [Pearl, 1988, Darwiche, 2009, Dechter, 2013], can be defined by a 3-tuple $\mathcal{M} = (\mathbf{X}, \mathbf{D}, \mathbf{F})$, where $\mathbf{X} = \{X_i : i \in V\}$ is a set of variables indexed by a set $V$ and $\mathbf{D} = \{D_i : i \in D\}$ is the set of finite domains of values for each $X_i$. Each function $f_\alpha \in \mathbf{F}$ is defined over a subset of the variables called its scope, $X_\alpha$, where $\alpha \subseteq V$ are the indices of variables in its scope and $D_\alpha$ denotes the Cartesian product of their domains, so that $f_\alpha : D_\alpha \to R^{\geq 0}$. **Primal graph** $\mathcal{G} = (\mathbf{V}, \mathbf{E})$ of $\mathcal{M}$ associates each variable with a node ($\mathbf{V} = \mathbf{X}$), while arcs $e \in \mathbf{E}$ connect nodes whose variables appear in the scope of the same function. $\mathcal{M}$ defines a global function, often a factorized probability distribution on $\mathbf{X}$, $P(\mathbf{X}) = \frac{1}{Z} \prod_\alpha f_\alpha(X_\alpha)$, where $Z$, known as the partition function, is a normalization factor.

Three common paradigms for solving inference tasks over graphical models are inference, search, and sampling. In our work we use inference for generation of a bounded heuristic to guide search for solving the K*MAP query. We use AND/OR search which utilizes an AND/OR space defined relative to a pseudo tree of the problem's primal graph generated with respect to a variable ordering. A **pseudo tree** $\mathcal{T} = (\mathbf{V}, \mathbf{E}')$ of a primal graph $\mathcal{G} = (\mathbf{V}, \mathbf{E})$ is a directed rooted tree that spans $\mathcal{G}$ with every arc of $\mathcal{G}$ not in $\mathbf{E}'$ as a back-arc in $\mathcal{T}$ connecting nodes to their ancestors (Figure 1(a),(b)). The pseudo tree structure determines the structure of the corresponding AND/OR search space. Its directed edges dictate how the search space should be explored with branchings in the pseudo tree corresponding to branchings emanating from the AND nodes in the search space allowing for conditional independencies to be exploited. For mixed inference problems where a subset of variables are to be maximized (MAP variables) and the remaining variables (SUM variables) marginalized, the pseudo tree must be constrained so that MAP variables precede SUM variables in the variable ordering [Marinescu et al., 2014].

Given a pseudo tree $\mathcal{T}$ of a primal graph $\mathcal{G}$, the AND/OR search tree $T_\mathcal{T}$ guided by $\mathcal{T}$ has alternating levels of OR nodes corresponding to variables, and AND nodes corresponding to assignments from their domains with edge costs extracted from the original functions [Dechter and Mateescu, 2007]. Each arc into an AND node $n$ has a cost $c(n)$ defined to be the product of all factors $f_\alpha$ in $\mathcal{M}$ that are instantiated at $n$ but not before (see Dechter and Mateescu [2007]).

The size of the search space is exponential in the depth of the pseudo tree (rather than the number of variables) and therefore mixed inference tasks such as MMAP can be solved more efficiently by traversing the AND/OR search space whenever its depth is bounded sufficiently.

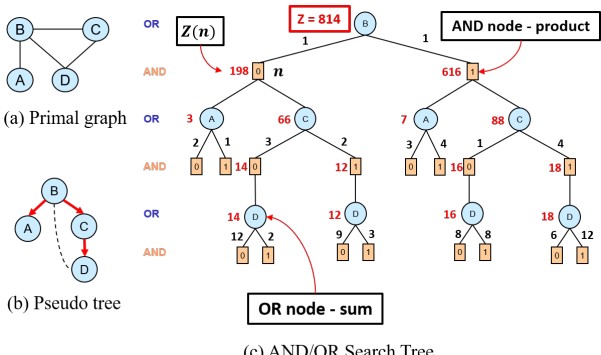

(a) Primal graph

(b) Pseudo tree

(c) AND/OR Search Tree

Figure 1: A full AND/OR tree representing all 16 solutions.

## 3 GRAPHICAL MODEL FOR K*MAP

As the first main contribution of this work, we describe two formulations of CPD problems as graphical models for computing K*MAP. These build upon previous work from MMAP (see Marinescu et al. [2018b]) and formulations for optimizing the weaker GMEC objective [Zhou et al., 2016].

### 3.1 FORMULATION 1 (F1)

**Variables and Domains:** Let $\boldsymbol{R} = \{R_i \mid i \in \{1, 2, ..., N\}\}$ be the set of **residue variables** representing $N$ different residues (i.e., positions) of the protein. Each $R_i$ has a corresponding domain $D_{R_i} = \{aa \mid aa \text{ is a possible amino acid assignment to residue } i\}$. For residues that are being considered for mutation (**mutable residues**), each $R_i$ considers one of ~20 possible amino acid assignments in its domain of values. These are the MAP variables maximized over in the K*MAP task.

$\boldsymbol{C_\gamma} = \{C_{\gamma(i)} \mid i \in \{1, 2, ..., N\}\}$ are **conformation variables**, each indexing a discrete spacial conformation (rotamer) of the amino acid at residue $R_i$ when its subunit is in form $\gamma$. Each $C_{\gamma(i)}$ has corresponding domain $D_{C_{\gamma(i)}} = \{1, 2, ..., M_{\gamma(i)}\}$, where $M_{\gamma(i)}$ is the maximum number of rotamers for any amino acid assignment to $R_i$ in form $\gamma$. The assignment to $C_{\gamma(i)}$ acts as an index to the possible side chain conformations of the amino acid assigned to $R_i$. The $\boldsymbol{C_\gamma}$ are the SUM variables to be marginalized over.

**Functions:** We consider two sets of functions: $E_\gamma^{sb} = \{E_{\gamma(i)}^{sb}(R_i, C_{\gamma(i)}) \mid i \in \{1, 2, ..., N\}\}$ that capture the energies of interaction of the amino acid at each residue $i$ with itself and the surrounding backbone, and $E_\gamma^{pw} = \{E_{\gamma(ij)}^{pw}(R_i, C_{\gamma(i)}, R_j, C_{\gamma(j)}) \mid \text{for } i, j \text{ s.t. } R_i \text{ and } R_j \text{ interact}\}$ that capture pair-wise energies of interaction between amino acids in close proximity. For any assignment to $C_{\gamma(i)}$ (which corresponds to a rotamer-index) that is out of range of $R_i$'s assigned amino acid's possible rotamers, an infinite energy value is assigned as an implicit constraint.

**Objective:** The K* objective can thus be expressed as

$K^*(R_1...R_N) = \frac{Z_B(R_1...R_N)}{Z_U(R_1...R_N)}$, where we assume temperature $T$ in Kelvin and Universal Gas Constant $\mathscr{R}$ and where:

$$Z_\gamma(R_1...R_N) = \sum_{C_{\gamma(1)},...,C_{\gamma(N)}} \prod_{E_{\gamma(i)}^{sb} \in \boldsymbol{E}_\gamma^{sb}} e^{-\frac{E_{\gamma(i)}^{sb}(R_i, C_{\gamma(i)})}{\mathscr{R}T}}$$
$$\cdot \prod_{E_{\gamma(ij)}^{pw} \in \boldsymbol{E}_\gamma^{pw}} e^{-\frac{E_{\gamma(ij)}^{pw}(R_i, C_{\gamma(i)}, R_j, C_{\gamma(j)})}{\mathscr{R}T}}$$

### 3.2 FORMULATION 2 (F2)

Formulation 2, inspired by the works of Viricel et al. [2018] and Vucinic et al. [2019], uses explicit constraints to restrict invalid amino acid - rotamer combinations.

**Variables and Domains:** As in F1, a set of residue variables $\boldsymbol{R} = \{R_i \mid i \in \{1, 2, ..., N\}\}$ and conformation variables $\boldsymbol{C_\gamma} = \{C_{\gamma(i)} \mid i \in \{1, 2, ..., N\}\}$ are considered. Here, each $C_{\gamma(i)}$ represents a specific amino acid and conformation of the $N$ different residues. Namely, each $C_{\gamma(i)}$ has a domain $D_{C_{\gamma(i)}} = \{c \mid c \text{ is a rotamer for one of the possible amino acids of } R_i\}$. The amino acid assignment to $R_i$ acts as a selector into the possible assignments to $C_{\gamma(i)}$.

**Functions:** $\mathscr{C} = \{\mathscr{C}_{\gamma(i)}(R_i, C_{\gamma(i)}) \mid i \in \{1, 2, ..., N\}, \gamma \in \varphi\}$ is a set of constraints ensuring that the assigned rotamer to $C_{\gamma(i)}$ belongs to the amino acid assigned to $R_i$. In addition, $E_\gamma^{sb} = \{E_{\gamma(i)}^{sb}(C_{\gamma(i)}) \mid i \in \{1, 2, ..., N\}\}$ captures the energies of interaction of the amino acid at each residue $i$ with itself and the surrounding backbone and $E_\gamma^{pw} = \{E_{\gamma(ij)}^{pw}(C_{\gamma(i)}, C_{\gamma(j)}) \mid \text{for } i, j \text{ s.t. } R_i \text{ and } R_j \text{ interact}\}$ captures the pair-wise energies of interaction between amino acids in close proximity.

**Objective:** Similar to before, the $K^*$ objective can be expressed as $K^*(R_1...R_N) = \frac{Z_B(R_1...R_N)}{Z_U(R_1...R_N)}$, where

$$Z_\gamma(R_1...R_N) = \sum_{C_1,...,C_N} \prod_{\mathscr{C}_{\gamma(i)} \in \mathscr{C}} \mathscr{C}_{\gamma(i)}(R_i, C_{\gamma(i)})$$
$$\cdot \prod_{E_{\gamma(i)}^{sb} \in \boldsymbol{E}_\gamma^{sb}} e^{-\frac{E_{\gamma(i)}^{sb}(C_{\gamma(i)})}{\mathscr{R}T}} \cdot \prod_{E_{\gamma(ij)}^{pw} \in \boldsymbol{E}_\gamma^{pw}} e^{-\frac{E_{\gamma(ij)}^{pw}(C_{\gamma(i)}, C_{\gamma(j)})}{\mathscr{R}T}}$$

With a graphical model framework in place, next we describe a heuristic that can be used to bound the $K^*$MAP value and that can be used to guide search.

## 4 WMBE-K*

Next we present a weighted mini-bucket based approximation scheme for $K^*$, which adapts weighted mini-bucket for MMAP [Dechter and Rish, 2002, Marinescu et al., 2014] to the new $K^*$ objective. Since the constrained-order mini-bucket bounds are compatible with AND/OR search, we will leverage these bounds in the subsequent section to guide search as part of a branch-and-bound scheme.

---

**Algorithm 1:** wMBE-K*

**input** : Graphical model $\boldsymbol{\mathcal{M}} = \{\boldsymbol{X}, \boldsymbol{D}, \boldsymbol{F}\}$; constrained variable order $o = [X_1, ..., X_n]$ (where $\boldsymbol{X} = \boldsymbol{R} \cup \boldsymbol{C_B} \cup \boldsymbol{C_U}$), i-bound $i$

**output** : upper bound on the $K^*$MAP value

1 **begin**

2    Partition the functions $f \in \boldsymbol{F}$ into buckets $B_n, ..., B_1$ s.t. each function is placed in the bucket corresponding to the highest-index variable in its scope.

3    **foreach** $k = n...1$ **do**

4      Generate a mini-bucket partitioning of the bucket functions $\boldsymbol{MB_k} = \{MB_k^1, ..., MB_k^T\}$ s.t. $vars(MB_k^t) \leq i$, for all $MB_k^t \in \boldsymbol{MB_k}$

5      **if** $X_k \in \boldsymbol{MAP}$ **then**

6        **foreach** $MB_k^t \in \boldsymbol{MB_k}$ **do**

7          $\lambda_k^t \leftarrow \max_{X_k} \prod_{f \in MB_k^t} f$

8      **else**

9        **if** $X_k \in \boldsymbol{C_B}$ **then**    // upper-bound for numerator

10          Select positive weights $\boldsymbol{w} = \{w_1, ..., w_T\}$ s.t. $\sum_{w_t \in \boldsymbol{w}} w_t = 1$

11          **foreach** $MB_k^t \in \boldsymbol{MB_k}$ **do**

12            $\lambda_k^t \leftarrow (\sum_{X_k} \prod_{f \in MB_k^t} f^{1/w_t})^{w_t}$

13        **else if** $X_k \in \boldsymbol{C_U}$ **then** // lower-bound for denominator

14          Select a negative weight $w_1$ and positive weights $\boldsymbol{w} = \{w_2, ..., w_T\}$ s.t. $\sum_{w_t \in \boldsymbol{w}} w_t = 1$

15          **foreach** $MB_k^t \in \boldsymbol{MB_k}$ **do**

16            $\lambda_k^t \leftarrow (\sum_{X_k} \prod_{f \in MB_k^t} f^{1/w_t})^{w_t}$

17            **if** $scope(\lambda_k^t) \cap \boldsymbol{C_U} = \emptyset$ **then**

18              $\lambda_k^t \leftarrow 1/\lambda_k^t$

19      Add each $\lambda_k^t$ to the bucket of the highest-index variable in its scope.

20    **return** $\lambda_1$

---

Algorithm wMBE-K* is described in Algorithm 1 and operates similarly to wMBE-MMAP [Marinescu et al., 2014]. Two key similarities are that (1) it takes a variable ordering that constrains buckets of MAP variables to be processed last (line 3) for which maximization occurs, and (2) for any bucket that has a width larger than a provided i-bound, a bounded approximation is made by partitioning the bucket functions into $T$ mini-buckets (line 4) and taking the product of their power-sums over the bucket variable (lines 9-12, 13-16), leveraging Holder's Inequality [Hardy et al., 1988].

For $K^*$MAP, two key innovations are required: (1) buckets corresponding to variables in $\boldsymbol{C_U}$, whose marginal belongs to the denominator of the $K^*$ expression, are lower-bounded (to lead to an upper bound on $K^*$) by using a modification to Holder's inequality that incorporates negative weights [Liu and Ihler, 2011] (lines 13-16), and (2) when messages are passed from buckets corresponding to variables in $\boldsymbol{C_U}$ to that of $\boldsymbol{R}$, the messages are inverted to accommodate being part of the denominator (line 18). Although details

are omitted here, wMBE-K$^*$ can also employ cost shifting to tighten its bounds (see Flerova et al. [2011], Liu and Ihler [2011]). In our empirical evaluation cost-shifting is implemented as well. Finally, the complexity of the algorithm, which is parametrized by the i-bound $i$, is time and space exponential in $i$ only.

As can be expected, bounding a ratio of functions (as in the case for K$^*$) is particularly challenging, relying on both upper and lower bounds. Lower bounding of functions is particularly challenging. For larger problems and low i-bounds, this can often yield relatively weak bounds. We provide an improvement to help remedy this next.

**Domain-Partitioned MBE** Until now, we used a mini-bucket scheme blind to explicit hard constraints and consistency issues. This can be handicapping when lower-bounding since constraints are represented as zero's in functions and can cause premature deflation of lower bounds. In particular, in mini-bucket elimination [Dechter and Rish, 2003] where lower bounds are created via minimizing over function values, zeros in the functions being minimized will cause the resulting lower bound itself to drop to zero. However, in the CPD domain where functions represent protein energetics, satisfiable configurations corresponds to positive function values and thus can guarantee a positive lower bound by using the following simple remedy:

Let $X$, $Y$, and $Z$ be three variables and let $obj = \sum_X f(x,y) \cdot g(x,z)$, let $X' = \{x \in X | g(x,z) \neq 0\}$ be a set such that $\epsilon_{X'} = min_{x \in X'} g(x,z)$. Clearly, $\epsilon_{X'} > 0$ and therefore we can derive: $obj = \sum_{x \in X'} f(x,y) \cdot g(x,z) + \sum_{x \in X \setminus X'} f(x,y) \cdot g(x,z) = \sum_{x \in X'} f(x,y) \cdot g(x,z) \geq \epsilon_{X'} \cdot \sum_{x \in X'} f(x,y) > 0$, assuming $f(x,z)$ is not identically zero over $X'$. Mini-buckets can then be computed according to such domain-partitions to improve their bounds.

## 5 AOBB-K$^*$

We present the key algorithmic contribution of this work: **AOBB-K$^*$** (Algorithm 2), a depth-first AND/OR branch-and-bound scheme for solving K$^*$MAP. With state-of-the-art K$^*$ optimizers such as BBK$^*$ employing memory-intensive best-first search [Ojewole et al., 2018, Hallen et al., 2018], depth-first algorithms provide scheme linear in space allowing for solving problems unable to be solved by best-first methodologies due to memory [Zhou et al., 2016].

AOBB-K$^*$ traverses the underlying AND/OR search tree guided by the provided pseudo tree $\mathcal{T}$, expanding nodes in a depth-first manner (line 9), and pruning whenever any of three conditions are triggered: (1) the resulting variable assignments violate constraints encoded as infinite values in $\mathcal{M}$ (line 10), (2) a subunit-stability constraint (SSC – a constraint which enforces the partition function of each protein subunit, $Z_\gamma$, to be greater than a biologically-relevant threshold $S_\gamma$ [Ojewole et al., 2018]) is violated (line 12),

---

**Algorithm 2:** AOBB-K$^*$

---

**input** : CPD graphical model $\mathcal{M}$; pseudo-tree $\mathcal{T}$; K$^*$ upper-bounding heuristic function $h_{K^*}(.)$; $Z_\gamma$ upper-bounding heuristic function $h_{Z_\gamma}(.)$; and subunit stability threshold $S_\gamma$ for each subunit $\gamma$

**output** : $K^*MAP(\mathcal{M})$

1 **begin**
2    Encode deterministic relations in $\mathcal{M}$ into CNF
3    $\pi \leftarrow$ root OR node $s$
4    $ub_{K^*}(s) \leftarrow h_{K^*}(s)$
5    $lb_{K^*}(s) \leftarrow -inf$
6    $g(s) \leftarrow 1$
7    **foreach** $\gamma \in \varphi$ **do**
8      $UB_{Z_\gamma}(s) \leftarrow \prod_{m \in ch_{T_\gamma}(s)} h_{Z_\gamma}(m)$
9    **while** $n_X \leftarrow EXPAND(\pi)$ **do**
10      **if** $ConstraintPropagation(\pi) = false$ **then**
11        $PRUNE(\pi)$
12      **else if** $\exists \gamma \in \varphi$ s.t. $UB_{Z_\gamma}(n_X) < S_\gamma$ **then**
13        $PRUNE(\pi)$
14      **else if** $X \in \mathbf{R}$ **then**
15        **if** $\exists a \in anc^{OR}(n)$ s.t. $ub_{K^*}(a,\pi) < lb_{K^*}(a)$ **then**
16          $PRUNE(\pi)$
17      **else if** $ch_T^{unexp}(n) = \emptyset$ **then**
18        $BACKTRACK(\pi)$
19    **return** $ub_{K^*}(s) = lb_{K^*}(s) = K^*MAP(\mathcal{M})$

---

or (3) it can be asserted that the current amino acid configuration cannot produce a K$^*$ better than any previously found (line 15). Backtracking occurs when all of a node's children have been explored and returned from (line 17), at which point the K$^*$ value of the sub problem the node roots is known exactly and bounds of its parents are tightened accordingly. Infinite energy tuples in $\mathcal{M}$'s functions model an inconsistent amino acid - rotamer pairs and correspond to hard constraints encoded into a CNF formula as in [Allen and Darwiche, 2003]. During search, unit-propagation (e.g., Eén and Sörensson [2004]) is propagates these constraints and detects infeasible variable configurations (line 10).

The algorithm progresses in this manner until it finally returns to, and updates, the root of the search tree with the maximal K$^*$ value corresponding to an amino acid configuration that also satisfies the subunit-stability thresholds.

Throughout search, each node $n$ maintains a progressive upper bound $ub_{K^*}(n)$ on the K$^*$MAP of the sub problem it roots. When a node is expanded, this value is initialized based on upper-bounding heuristic function $h_{K^*}^{ub}(.)$ (line 4). As search progresses, $ub_{K^*}(n)$ decreases converging towards the K$^*$MAP of the sub problem rooted at $n$. Furthermore, each node $n$ also maintains a progressively improved upper bound on the partition function of each subunit $\gamma$ consistent with the path to $n$, $UB_{Z_\gamma}(n)$ (line 8). At each step in the search, $UB_{Z_\gamma}(n)$ is recomputed to ensure that it is greater than the given $S_\gamma$, thus satisfying the SSC's

and enforcing consideration of only biologically relevant solutions [Ojewole et al., 2018]. Note that the SSC's are not encoded into the problem, and thus add another layer of complexity not present in tasks such as MMAP.

**Theorem 5.1** (correctness, completeness). *AOBB-K\* is sound and complete, returning the optimal $K^*$ value of all amino-acid configurations that do not violate the subunit-stability constraints.*

**Theorem 5.2** (complexity). *AOBB-K\* is time $O(n \cdot k^d)$, where $n$ is the number of variables, $k$ is the maximum domain size, and $d$ is the dept of the guiding pseudo tree, and is space $O(n)$ (see Marinescu and Dechter [2009b]). When modified to search the context minimal AND/OR search graph (as opposed to search tree) AOBB-K\* is both time and space $O(n \cdot k^{w^*})$, where $w^*$ is the induced width of the pseudo tree (see Marinescu and Dechter [2009a]).*

## 5.1 WEIGHTED SEARCH FOR K\*

Weighted best-first search (e.g., WA\* [Pohl, 1970], WAO\* [Desarkar et al., 1987]) is a well known principle for converting best-first search into an anytime scheme by multiplying the heuristic function $h(n)$ of a node $n$ in the search space by a weight $\omega > 1$. The solution is guaranteed to be $\omega$-optimal (i.e., within a factor $\omega$ from the optimal one).

Therefore, AOBB-K\* can easily be relaxed to an $\omega$-approximation scheme (for $\omega \in [0, 1)$) by multiplying $h_{K^*}(n)$ at each node $n$ in the AND/OR search tree by a factor of $\omega$. It can be shown that the resulting solution will be at worst $\omega \cdot K^* MAP$ [Pohl, 1970, Flerova et al., 2014]. We explore the performance of applying such approximations to a class of difficult CPD problem instances in Section 6.

## 5.2 INFUSING DETERMINISM VIA THRESHOLDED UNDERFLOWS

Exploiting determinism via constraint propagation (**CP**) can be a powerful tool [Dechter, 2003, 2019] helping to prune invalid configurations corresponding to inconsistent amino acid - rotamer pairs or configurations not contributing to a subunit's partition function. CP can be accelerated by introducing additional determinism such that the solution is unchanged. Namely, extremely small function values (corresponding to extremely unfavorable side-chain interactions that would not occur in feasible solutions) can be underflowed (ie. replaced by zeros) and treated as hard constraints.

**Definition 5.1** ($\tau$-underflow of $f$, $f^\tau$). *Let $f$ be a non-negative function and $\tau \in \mathbb{R}^+$. The $\tau$-underflow of $f$ is $f^\tau(x) = f(x)$ if $f(x) \geq \tau$ and 0, otherwise.*

**Definition 5.2** ($\tau$-underflow of $\mathcal{M}$, $\mathcal{M}^\tau$). *For $\mathcal{M} = \langle \mathbf{X}, \mathbf{D}, \mathbf{F} \rangle$, the $\tau$-underflow of $\mathcal{M}$ is $\mathcal{M}^\tau = \langle \mathbf{X}, \mathbf{D}, \mathbf{F}^\tau \rangle$, where $\mathbf{F}^\tau = \{f^\tau \mid f \in \mathbf{F}\}$.*

**Notation.** We denote: $f_\gamma^{max}$ - the maximum value in all the functions associated with subunit $\gamma$, $|\boldsymbol{f_\gamma}|$ - the number of functions included in the partition function computation for subunit $\gamma$, excluding explicit constraints, $|\boldsymbol{C_\gamma}|^\Uparrow$ - the cardinality of the Cartesian product of greatest number of assignments to each variable in $C_\gamma$ that are individually consistent with any assignment $\boldsymbol{R} = r$, $Z_B^{min}$ - the smallest $Z_B$ that could possibly lead to a valid $K^* > K^{*(wt)}$ (here $wt$ refers to the **wild-type**, or naturally occurring, amino acid sequence), $prec(x)$ - the smallest decimal place for which the value of $x$ is recorded.

We give two sufficient conditions that, when upheld, allow us to assert $K^*MAP(\mathcal{M}^\tau) = K^*MAP(\mathcal{M})$: (1) $\tau$-underflows do not alter subunit partition functions that satisfy SSC's, i.e., $\forall r \in \boldsymbol{R}$ s.t. $Z_\gamma^{\mathcal{M}}(r) \geq S_\gamma$, $Z_\gamma^{\mathcal{M}}(r) = Z_\gamma^{\mathcal{M}^\tau}(r)$ and (2) $\tau$-underflows do not alter any $Z_B \geq Z_B^{min}$, i.e., $\forall r \in \boldsymbol{R}$ s.t. $Z_B^{\mathcal{M}}(r) \geq Z_B^{min}$, $Z_B^{\mathcal{M}}(r) = Z_B^{\mathcal{M}^\tau}(r)$.

**Condition 1:** A problem created with underflows using an underflow threshold $\tau$ such that $\forall \gamma \in \varphi$: (i) if $\tau \leq 1$, $|\boldsymbol{C_\gamma}|^\Uparrow \cdot (f_\gamma^{max})^{|\boldsymbol{f_\gamma}|} \cdot (\tau) < 0.5 \cdot prec(S_\gamma)$, or (ii) if $\tau > 1$, $|\boldsymbol{C_\gamma}|^\Uparrow \cdot (f_\gamma^{max})^{|\boldsymbol{f_\gamma}|} \cdot (\tau)^{|\boldsymbol{f_\gamma}|} < 0.5 \cdot prec(S_\gamma)$ will not violate any subunit stability constraints.

**Condition 2:** A problem created with underflows using an underflow threshold $\tau$ such that $\tau$ does not violate the SSC's and such that: (i) if $\tau \leq 1$, $|\boldsymbol{C_B}|^\Uparrow \cdot (f_B^{max})^{|\boldsymbol{f_B}|} \cdot (\tau) < 0.5 \cdot prec(Z_B^{min})$, or (ii) if $\tau > 1$, $|\boldsymbol{C_B}|^\Uparrow \cdot (f_B^{max})^{|\boldsymbol{f_B}|} \cdot (\tau)^{|\boldsymbol{f_B}|} < 0.5 \cdot prec(Z_B^{min})$, will also not alter the $K^*MAP$ solution.

**Theorem 5.3.** *If a chosen $\tau$ satisfies Conditions 1 and 2, then $K^*MAP(\mathcal{M}^\tau) = K^*MAP(\mathcal{M})$.*

**Complexity.** Both Conditions 1 and 2 can be verified in linear time, thus the sufficient condition presented in Theorem 5.3 can also be checked in linear time.

Although Theorem 5.3 provides a way to verify a $\tau$ used will still result in the correct optimal solution, the theorem's underlying conditions are often overly conservative, and, in practice, even $\tau$ larger than those satisfying the theorem can yield exact solutions.

# 6 EMPIRICAL EVALUATION

We now present the empirical evaluation of our schemes on real protein benchmarks provided to us by the Bruce Donald Lab at Duke University.

**Benchmarks.** We experimented on 48 protein design benchmarks, 30 with two mutable residues (denoted "original"),

12 made harder expanding to three mutable residues ("expanded"), and six problems expanded to four mutable residues ("hard"). Problems were generated using OSPREY 3.0 [Hallen et al., 2018] to compute interaction energies and then formulated into F1 and F2.

**AOBB-K\* vs state-of-the-art BBK\*.** We ran AOBB-K\* and its derived schemes, $[\omega, \tau]$-AOBB-K\*, built on top of AOBB-MMAP [Marinescu et al., 2014] implemented in C++. For comparison, we also run BBK\* [Ojewole et al., 2018], a state-of-the-art best-first search algorithm available in OSPREY 3.0. BBK\* (implemented in Java, set to use rigid rotamers and with a bound-tightness of $1 \times 10^{-2001}$).

Contrasting major differences of the two schemes, (1) AOBB-K\* uses a depth-first traversal allowing for memory efficiency and ease of facilitating constraint propagation while BBK\* uses memory-intensive best-first search, (2) our experiments with AOBB-K\* use the statically compiled wMBE-K\* generated before search while BBK\* exclusively uses a dynamic heuristic that considers best and worst case scenarios from all possible ensuing configurations during search, (3) we use constraint propagation dynamically during search to prune provably invalid (or zero-weighted) regions of the search space, and (4) AOBB-K\* is designed as an exact scheme that finds the optimal K\* that satisfies SSC's (ie. exact for the K\*MAP task) whereas BBK\* is inherently designed as a bounded approximation scheme that, even when using the very low tightness of $1 \times 10^{-200}$, seems not to always return the optimal K\*MAP.

Experiments were allotted 1hr on a 2.66 GHz processor with 4 GB of memory and using subunit-stability thresholds $S_\gamma = Z_\gamma^{(wt)} \cdot e^{-\frac{5}{\mathscr{R}T}}$ where $Z_\gamma^{(wt)}$ is the partition function of the wild-type sequence. As BBK\* can take advantage of parallelism, it was run with access to 4 CPU cores.

AOBB-K\* was run using wMBE-K\* with moment matching [Flerova et al., 2011, Liu and Ihler, 2011] for guiding and bounding search. For hard problems, a derivation of wMBE-K\*, wMBE⁺-K\*, was tested that avoided consideration of zeros during lower-bounding approximations (motivated by Section 4). For all experiments, wMBE-MMAP was used to upper-bound the partition function of each subunit. According to Ojewole et al. [2018], BBK\* uses a dynamic greedy heuristic based on optimistic values for all variables not yet instantiated.

In all problems, each mutable residues considers 21 different amino acid assignments. Conformation variables of non-mutable residues have a domain size of 2-14 rotamers (most having domain sizes 4-9). Conformation variables of mutable residues have a domain size of 34-35 when formulated as F1 and 203-205 when formulated as F2.

**Table Keys. F**: formulation type, $\omega$: weight applied to the

---

¹BBK\*'s bound tightness parameter does not correlate directly with an $\omega$-approximation. See Ojewole et al. [2018].

Table 1: Aggregated statistics comparing the solution and time of AOBB-K\* (on both F1 and F2, as an exact and anytime scheme) with BBK\*. **K\*≥** counts the the number of times AOBB-K\*'s K\* solution was equal or greater; **K\*>** counts the number of AOBB-K\*'s solution was strictly better; **time<** counts the number of times AOBB-K\* found the optimal solution faster.

| Dataset | AOBB-K\* | | | any-AOBB-K\* | |
|---|---|---|---|---|---|
| | K\*≥ | K\*> | time< | K\*≥ | K\*> |
| (#instances) | (F1,F2) | (F1,F2) | (F1,F2) | (F1,F2) | (F1,F2) |
| Orig. (30) | 30,30 | 2,2 | 23,28 | 30,30 | 2,2 |
| Expand. (12) | 6,11 | 0,4 | 2,4 | 11,11 | 1,4 |

heuristic, $\tau$: underflow-threshold, **H**: heuristic, **iB**: i-bound used, **w\***: induced width, **d**: pseudo tree depth, **|R|**: number of MAP variables, **|X|**: total number of variables, $Dmax$: maximum domain size, **UB**: heuristic upper bound at the root (empty cells representing no finite bound), **pre-t**: pre-processing time (ex. compiling heuristics), **search**: search time, **time**: total time, **K\***: the returned K\* solution (in $log_{10}$), and **Any-K\***: best valid K\* value found in 1hr. The best performance points are highlighted.

Table 2: F1 vs F2 on original and Expanded(*) problems.

| benchmark | F | iB | \|R\| | \|X\| | Dᵐᵃˣ | w\* | d | UB | pre-t | time |
|---|---|---|---|---|---|---|---|---|---|---|
| 1a0r_00031 | F1 | 3 | 2 | 16 | 34 | 8 | 8 | | 0.9 | 226.9 |
| | F2 | 3 | 2 | 16 | 203 | 6 | 8 | | 0.7 | **93.0** |
| 1gwc_00021 | F1 | 6 | 2 | 12 | 34 | 6 | 6 | 10.28 | 69.2 | 101.2 |
| | F2 | 4 | 2 | 12 | 203 | 4 | 6 | 10.29 | 6.8 | **15.7** |
| 1gwc_00033 | F1 | 3 | 2 | 18 | 35 | 9 | 9 | | 1.0 | 30.2 |
| | F2 | 3 | 2 | 18 | 203 | 7 | 9 | | 1.3 | **12.3** |
| 2hnu_00026 | F1 | 6 | 2 | 14 | 34 | 7 | 7 | 15.18 | 14.6 | 25.0 |
| | F2 | 4 | 2 | 14 | 203 | 5 | 7 | 15.08 | 1.7 | **7.3** |
| 2rfe_00012* | F1 | 3 | 3 | 15 | 34 | 8 | 8 | | 2.9 | **24.5** |
| | F2 | 4 | 3 | 15 | 205 | 5 | 8 | 14.80 | 58.3 | 60.9 |
| 2xgy_00020* | F1 | 6 | 3 | 15 | 35 | 8 | 8 | 12.28 | 122.0 | 775.2 |
| | F2 | 5 | 3 | 15 | 203 | 5 | 8 | 11.39 | 81.8 | **360.7** |
| 3u7y_00011* | F1 | 5 | 3 | 13 | 34 | 7 | 7 | | 12.9 | **45.1** |
| | F2 | 4 | 3 | 13 | 203 | 4 | 7 | 12.29 | 74.0 | 76.1 |
| 4wwi_00019* | F1 | 6 | 3 | 15 | 34 | 8 | 8 | 16.93 | 119.6 | 1494.9 |
| | F2 | 5 | 3 | 15 | 203 | 5 | 8 | 16.05 | 169.2 | **181.3** |

**Formulation 1 vs. Formulation 2:** We can see from the aggregated time statistics in Table 1 that F2 is generally superior to F1, outperforming BBK\* more frequently. F1 uses an indexing scheme between amino-acids and their rotamers, which implies that all functions need to include *both* residue and conformation variables thus leading to densely connected graphs compared with F2. A strength of F1 vs. F2 is its smaller domain sizes in the presence of MAP variables which facilitates use of higher $i$-bounds compared with F2.

**AOBB-K\*vs BBK\* (solution quality):** The summary in Table 1 show that AOBB-K\* and BBK\* find the same K\* solution for the majority of the original problems. As problems are expanded, AOBB-K\* begins to find better solutions more frequently. Concrete examples are displayed in Tables 3 and 4 where greater K\* solutions by AOBB-K\* are highlighted in the K\*MAP columns.

**AOBB-K\*vs BBK\* (running time):** The results in Table 3 show that AOBB-K\* solved nearly all original benchmark problems faster than BBK\* (see the **time** column). As problems were expanded (Table 4), we see AOBB-K\*'s time-performance begin to drop more rapidly than BBK\*'s; AOBB-K\* surpasses BBK\* on four problems, but not on the other eight. No hard problem was solved exactly without modifications (presented shortly). However, it is important to note that: (1) AOBB-K\* finds solutions greater than that of BBK\* (which may account for increased time searching) and not explicitly shown here (2) AOBB-K\* returns intermediate anytime solutions, some of which exceeded the K\* value returned by BBK\*.

Table 3: AOBB-K\* vs BBK\* on Original F2 problems.

| | | | | | | AOBB-K\* | | BBK\* | |
| benchmark | iB | w\* | \|X\| | UB | pre-t | (203 ≤ Dmax ≤ 206) time | K\* | time | K\* |
|---|---|---|---|---|---|---|---|---|---|
| 1a0r_00031 | 3 | 6 | 16 | | 0.7 | 92.99 | 7.88 | 109.1 | 7.88 |
| 1gwc_00021 | 4 | 4 | 12 | 10.29 | 6.8 | 15.68 | 9.79 | 152.3 | 9.79 |
| 1gwc_00033 | 3 | 7 | 18 | | 1.3 | 12.33 | 10.48 | 512.5 | 10.48 |
| 2hnu_00026 | 4 | 5 | 14 | 15.08 | 1.7 | 7.28 | 13.18 | 436.9 | 13.18 |
| 2hnv_00025 | 4 | 6 | 16 | 15.04 | 1.7 | 16.56 | 13.65 | 962.1 | 13.65 |
| 2rf9_00007 | 6 | 7 | 18 | 14.52 | 4.1 | 4.67 | 14.08 | 45.5 | 14.08 |
| 2rf9_00013 | 5 | 6 | 16 | 14.12 | 1.4 | 1.93 | 13.25 | 11.8 | 13.25 |
| 2rf9_00018 | 6 | 7 | 18 | 16.68 | 8.4 | 15.13 | 15.79 | 187.2 | 15.79 |
| 2rf9_00042 | 6 | 9 | 22 | | 15.8 | 148.78 | 22.65 | 897.1 | 22.65 |
| 2rfd_00035 | 6 | 6 | 16 | 17.70 | 80.1 | 379.82 | 17.27 | 1242.3 | 16.77 |
| 2rfe_00012 | 4 | 5 | 14 | 14.80 | 0.8 | 1.49 | 13.93 | 11.1 | 13.93 |
| 2rfe_00014 | 4 | 5 | 14 | 15.23 | 0.8 | 1.72 | 14.36 | 31.4 | 14.36 |
| 2rfe_00017 | 5 | 5 | 14 | 10.96 | 1.7 | 7.02 | 10.52 | 29.2 | 10.52 |
| 2rfe_00030 | 4 | 5 | 14 | 11.53 | 6.9 | 18.94 | 10.50 | 181.5 | 10.50 |
| 2rfe_00041 | 5 | 7 | 18 | | 48.1 | 401.68 | 22.73 | 1181.5 | 22.73 |
| 2rfe_00043 | 6 | 6 | 16 | 18.48 | 75.5 | 79.4 | 18.04 | 50.5 | 18.04 |
| 2rfe_00044 | 6 | 6 | 16 | 18.62 | 75.2 | 85.6 | 18.19 | 74.5 | 18.19 |
| 2rfe_00047 | 3 | 7 | 18 | | 0.7 | 88.27 | 22.70 | 348.9 | 22.70 |
| 2rfe_00048 | 4 | 8 | 20 | | 2.8 | 158.65 | 22.81 | 387.7 | 22.81 |
| 2rl0_00008 | 4 | 3 | 10 | 11.16 | 2.6 | 2.62 | 11.16 | 261.9 | 9.46 |
| 2xgy_00020 | 4 | 5 | 14 | 11.47 | 1.9 | 16.36 | 10.60 | 887.5 | 10.60 |
| 3cal_00032 | 6 | 6 | 16 | 13.38 | 59.5 | 125.26 | 11.62 | 1428.9 | 11.62 |
| 3ma2_00016 | 4 | 5 | 14 | 13.39 | 0.7 | 3.51 | 8.38 | 9.6 | 8.38 |
| 3u7y_00009 | 6 | 4 | 12 | 4.51 | 5.7 | 5.72 | 4.51 | 190.8 | 4.51 |
| 3u7y_00011 | 3 | 4 | 12 | 12.72 | 0.5 | 1.95 | 11.85 | 27.8 | 11.85 |
| 4hem_00027 | 3 | 6 | 16 | | 1.0 | 2.01 | 15.48 | 39.9 | 15.48 |
| 4hem_00028 | 3 | 6 | 16 | | 0.9 | 1.84 | 15.27 | 34.5 | 15.27 |
| 4kt6_00023 | 4 | 6 | 16 | 14.80 | 2.1 | 7.30 | 12.69 | 136.5 | 12.69 |
| 4kt6_00024 | 4 | 6 | 16 | 14.87 | 2.1 | 5.07 | 12.93 | 120.7 | 12.93 |
| 4wwi_00019 | 5 | 5 | 14 | 15.43 | 5.6 | 7.85 | 14.99 | 26.3 | 14.99 |

Table 4: AOBB-K\* vs BBK\* on Expanded F2 problems.

| | | | | | | AOBB-K\* | | BBK\* | |
| benchmark | iB | w\* | \|X\| | UB | pre-t | (203 ≤ Dmax ≤ 206) time | K\* | time | K\* |
|---|---|---|---|---|---|---|---|---|---|
| 1gwc_00021\* | 4 | 4 | 13 | 12.51 | 123.8 | 205.1 | 11.92 | 551.3 | 11.72 |
| 2hnv_00025\* | 4 | 6 | 17 | 18.38 | 109.8 | 153.8 | 16.18 | 880.5 | 13.65 |
| 2rf9_00007\* | - | - | - | - | - | - | - | 369.4 | 14.73 |
| 2rf9_00013\* | 4 | 6 | 17 | 16.36 | 83.0 | 100.8 | 15.03 | 39.2 | 15.03 |
| 2rfe_00012\* | 4 | 5 | 15 | 14.80 | 58.3 | 60.9 | 13.93 | 11.8 | 13.93 |
| 2rfe_00014\* | 4 | 5 | 15 | 15.23 | 58.2 | 60.6 | 14.36 | 44.9 | 14.36 |
| 2rfe_00017\* | 5 | 5 | 15 | 11.46 | 166.8 | 334.6 | 10.86 | 78.0 | 10.80 |
| 2rfe_00030\* | 4 | 5 | 15 | 13.61 | 115.2 | 276.6 | 11.12 | 275.4 | 10.97 |
| 2xgy_00020\* | 5 | 5 | 15 | 11.39 | 81.8 | 360.7 | 10.96 | 1388.1 | 10.96 |
| 3u7y_00009\* | 4 | 4 | 13 | 4.95 | 62.6 | 99.5 | 4.51 | 215.8 | 4.51 |
| 3u7y_00011\* | 4 | 4 | 13 | 12.29 | 74.0 | 76.1 | 11.85 | 26.6 | 11.85 |
| 4wwi_00019\* | 5 | 5 | 15 | 16.05 | 169.2 | 181.3 | 14.99 | 34.0 | 14.99 |

**Weighted Search:** Each pair of rows in Table 5 compares AOBB-K\*(denoted $\omega = 1$) and $\omega$-AOBB-K\*($\omega =$

0.001) on all Expanded problems. Moving to approximate search reduces the MAP search space and improves search time (sometimes by more than a factor of ten). For 2rfe_prepped_00017\*, 2xgy_prepped_00020\*, and 4wwi_prepped_00019\*, we see the optimal i-bound is reduced from 5 to 4 when using weighted search highlighting the increased pruning of the search space enabled by the weighted heuristic even when using a weaker (but more quickly computed) heuristic. (When using the same i-bound, the weighted search still showed significant speed-ups solving these problems). With other advances, weighted search was instrumental to solving the hard problems.

**Infusing Determinism:** We assessed the impact of infusing determinism as discussed in Section 5.2 by applying thresholded-underflow with $\tau = 1 \times 10^{-5}$ and comparing to base problems ($\tau = 0$). Table 6 shows the results obtained on expanded problems. In all cases, we see that the $\tau$-underflows improved search times (see "search" column), sometimes so much so that the best time corresponded to allowing for a more crudely computed heuristic to enter search more quickly. Even when the same iB is used, the overall time is still reduced, however interestingly the speedup does not always correspond to a reduced search of the MAP space; we found that the heuristic can be adversely affected by underflows (as described in Section 4's discussion on Domain-Partitioned MBE) resulting in less efficient pruning of the MAP space. However, this can potentially be remedied, which we explore next.

**wMBE$^+$-K\*:** Bounding a ratio of functions such as in K\* is especially difficult - particularly because it relies on lower-bounding, which is especially problematic in the presence of determinism. To empirically test the intuition presented in Section 4 and in attempt to solve problems that could not be solved by vanilla AOBB-K\*, we used wMBE$^+$-K\* (which avoids considering zeros when lower bounding) with $\omega, \tau$-AOBB-K\* ($\omega = 0.001, \tau = 1 \times 10^{-5}$) on hard F1 problems, for which wMBE-K\* was unbounded (Table 7). In every case, the modified heuristic was able to provide a bounded estimate and in three of the six problems enabling AOBB-K\* to find a solution better than BBK\*.

**In summary**, AOBB-K\* shows strong performance on the original protein problems, especially with Formulation F2, outperforming state-of-the-art BBK\* time-wise and finding better solutions. AOBB-K\* also performed admirably on expanded problems, finding solutions faster for several problems and finding better solutions. The innovations of $\omega$-approximate search, infusing determinism, and omitting zeros from lower bounding enabled the algorithm to find solutions to harder problems, some of which were better than BBK\*'s. However, these hard problems were a challenge for base AOBB-K\*. The increase in domain sizes as MAP variables increase proved problematic memory-wise for compiling the heuristic, restricting to low iB's resulting in less helpful bounds. This challenge, and the realization

that bounds can be improved by domain partitioning, motivates future work into new representations.

# 7 SUMMARY, CONCLUSION, AND FOUNDATION FOR FUTURE WORK

**Summary.** This work creates a competitive framework for which future advancements to protein redesign optimizing $K^*$ can be made leveraging advances in graphical model algorithms. This work also features an exploration of new bounded heuristics for the $K^*$MAP task and identifies areas amenable to improvement. As part of the presented framework, new efficient algorithms solving the $K^*$MAP task are introduced along with an empirical proof of concept by evaluation against a pre-existing state-of-the-art algorithm.

**Conclusion.** We leave readers with **1.** two distinct graphical model formulations to address the $K^*$MAP query, and analysis showing Formulation F2 is generally superior; **2.** a new guiding wMBE-$K^*$ heuristic and demonstration of the potential to improve its bounds by domain-partitioning; **3.** AOBB-$K^*$, a depth-first AND/OR branch-and-bound algorithm for optimizing $K^*$ (and an accompanying approximate $\omega$-AOBB-$K^*$) that proved competitive against state-of-the-art BBK$^*$; **4.** a scheme to exploit determinism into problems by introducing underflows, with theoretical guarantees; and **5.** extensive empirical analysis on over forty benchmarks providing the above-mentioned analysis. As a part of our results, we note that AOBB-$K^*$ finds a $K^*$ greater than that of BBK$^*$ on several occasions, leaving an open question as to the cause and how it impacts time-performance.

**As a foundation.** Research directions illuminated by our results are: **(1)** exploration of more compact representations able to exploit determinism [Mateescu and Dechter, 2008, Larkin and Dechter, 2003]; **(2)** development of new $K^*$ heuristics (incorporating constraints, using alternative frameworks such as Deep Bucket Elimination [Razeghi et al., 2021]), employing dynamic heuristics); and **(3)** adaptation of state-of-the-art mixed-inference schemes to $K^*$MAP [Lou et al., 2018a,b, Marinescu et al., 2019, 2018a,b]. This work can be tuned more to the protein domain by: **(1)** evaluating new CPD's with additional independencies exploitable by AND/OR schemes, **(2)** integrating optimizations present in developed protein software such as BBK$^*$'s OSPREY 3.0; and **(3)** considering backbone ensembles.

### Acknowledgements

Special thanks to Thomas Schiex, from INRAE, France, and Bruce Donald and his members of his lab at Duke University, including Graham Holt, Jonathan Jou, and Nathan Guerin. We also thank our reviewers for their valuable comments. This work was supported in part by NSF grant IIS-2008516.

Table 5: AOBB-$K^*$ ($\omega = 1$) vs. Weighted $\omega$-AOBB-$K^*$ ($\omega = 0.001$) on Expanded F2 problems.

| benchmark | $\omega$ | iB | w* | d | |X| | UB | pre-t | search | time | K* |
|---|---|---|---|---|---|---|---|---|---|---|---|
| | | | | | | (203 ≤ Dmax ≤ 206) | | | $\omega$-AOBB-K* | |
| 1gwc_00021* | 1 | 4 | 4 | 7 | 13 | 28.80 | 123.8 | 81.3 | 205.1 | 11.92 |
| | 0.001 | 4 | 4 | 7 | 13 | 28.80 | 124.3 | 12.1 | 136.4 | |
| 2hnv_00025* | 1 | 4 | 6 | 9 | 17 | 42.32 | 109.8 | 44.0 | 153.8 | 16.18 |
| | 0.001 | 4 | 6 | 9 | 17 | 42.32 | 109.3 | 12.2 | 121.5 | |
| 2rf9_00013* | 1 | 4 | 6 | 9 | 17 | 37.68 | 83.0 | 17.8 | 100.8 | 15.03 |
| | 0.001 | 4 | 6 | 9 | 17 | 37.68 | 82.8 | 1.6 | 84.4 | |
| 2rfe_00012* | 1 | 4 | 5 | 8 | 15 | 34.07 | 58.3 | 2.6 | 60.9 | 13.93 |
| | 0.001 | 4 | 5 | 8 | 15 | 34.07 | 58.6 | 0.3 | 58.9 | |
| 2rfe_00014* | 1 | 4 | 5 | 8 | 15 | 35.07 | 58.2 | 2.4 | 60.6 | 14.36 |
| | 0.001 | 4 | 5 | 8 | 15 | 35.07 | 58.2 | 1.3 | 59.4 | |
| 2rfe_00017* | 1 | 5 | 5 | 8 | 15 | 26.39 | 166.8 | 167.8 | 334.6 | 10.86 |
| | 0.001 | 4 | 5 | 8 | 15 | 27.39 | 89.1 | 5.0 | 94.1 | |
| 2rfe_00030* | 1 | 4 | 5 | 8 | 15 | 31.34 | 115.2 | 161.5 | 276.6 | 11.1 |
| | 0.001 | 4 | 5 | 8 | 15 | 31.34 | 101.4 | 0.5 | 101.9 | 10.9 |
| 2xgy_00020* | 1 | 5 | 5 | 8 | 15 | 26.24 | 81.8 | 278.9 | 360.7 | 10.96 |
| | 0.001 | 4 | 5 | 8 | 15 | 27.24 | 60.4 | 8.2 | 68.6 | |
| 3u7y_00009* | 1 | 4 | 4 | 7 | 13 | 11.41 | 62.6 | 36.8 | 99.5 | 4.51 |
| | 0.001 | 4 | 4 | 7 | 13 | 11.41 | 62.5 | 2.1 | 64.7 | |
| 3u7y_00011* | 1 | 4 | 4 | 7 | 13 | 28.29 | 74.0 | 2.1 | 76.1 | 11.85 |
| | 0.001 | 4 | 4 | 7 | 13 | 28.29 | 83.4 | 0.1 | 83.5 | |
| 4wwi_00019* | 1 | 5 | 5 | 8 | 15 | 36.96 | 169.2 | 12.1 | 181.3 | 14.99 |
| | 0.001 | 4 | 5 | 8 | 15 | 37.96 | 62.0 | 7.2 | 69.2 | |

Table 6: Impact of $\tau$-underflows on Expanded F2 problems.

| benchmark | $\tau$ | iB | w* | d | |X| | UB | pre-t | search | time | K* |
|---|---|---|---|---|---|---|---|---|---|---|---|
| | | | | | | (203 ≤ Dmax ≤ 206) | | | $\omega$-AOBB-K* | |
| 1gwc_00021* | 0 | 4 | 4 | 7 | 13 | 28.80 | 123.8 | 81.3 | 205.1 | 11.92 |
| | 1E-05 | 4 | 4 | 7 | 13 | 28.80 | 117.1 | 3.5 | 120.7 | |
| 2hnv_00025* | 0 | 4 | 6 | 9 | 17 | 42.32 | 109.8 | 44.0 | 153.8 | 16.18 |
| | 1E-05 | 4 | 6 | 9 | 17 | 42.32 | 100.8 | 1.7 | 102.4 | |
| 2rf9_00013* | 0 | 4 | 6 | 9 | 17 | 37.68 | 83.0 | 17.8 | 100.8 | 15.03 |
| | 1E-05 | 4 | 6 | 9 | 17 | 37.68 | 71.4 | 0.4 | 71.8 | |
| 2rfe_00012* | 0 | 4 | 5 | 8 | 15 | 34.07 | 58.3 | 2.6 | 60.9 | 13.93 |
| | 1E-05 | 3 | 5 | 8 | 15 | | 5.5 | 14.9 | 20.3 | |
| 2rfe_00014* | 0 | 4 | 5 | 8 | 15 | 35.07 | 58.2 | 2.4 | 60.6 | 14.36 |
| | 1E-05 | 3 | 5 | 8 | 15 | | 4.9 | 15.2 | 20.0 | |
| 2rfe_00017* | 0 | 5 | 5 | 8 | 15 | 26.39 | 166.8 | 167.8 | 334.6 | 10.86 |
| | 1E-05 | 4 | 5 | 8 | 15 | 27.39 | 85.7 | 7.1 | 92.8 | |
| 2rfe_00030* | 0 | 4 | 5 | 8 | 15 | 31.34 | 115.2 | 161.5 | 276.6 | 11.12 |
| | 1E-05 | 4 | 5 | 8 | 15 | 30.43 | 105.4 | 3.5 | 108.9 | |
| 2xgy_00020* | 0 | 5 | 5 | 8 | 15 | 26.24 | 81.8 | 278.9 | 360.7 | 10.96 |
| | 1E-05 | 4 | 5 | 8 | 15 | 27.24 | 60.1 | 23.8 | 83.9 | |
| 3u7y_00009* | 0 | 4 | 4 | 7 | 13 | 11.41 | 62.6 | 36.8 | 99.5 | 4.51 |
| | 1E-05 | 4 | 4 | 7 | 13 | | 61.3 | 5.1 | 66.4 | |
| 3u7y_00011* | 0 | 4 | 4 | 7 | 13 | 28.29 | 74.0 | 2.1 | 76.1 | 11.85 |
| | 1E-05 | 4 | 4 | 7 | 13 | 28.29 | 80.7 | 0.5 | 81.2 | |
| 4wwi_00019* | 0 | 5 | 5 | 8 | 15 | 36.96 | 169.2 | 12.1 | 181.3 | 14.99 |
| | 1E-05 | 4 | 5 | 8 | 15 | | 54.2 | 8.4 | 62.7 | |

Table 7: Observing the effects on wMBE approximations on Hard problems when omitting zeros while computing lower bounds.

| benchmark | H | iB | w* | |X| | UB | $\omega$-AOBB-K* ($\omega$=0.001) time | K* | BBK* time | K* |
|---|---|---|---|---|---|---|---|---|---|
| (τ = 1e-5) | | | | | | (34 ≤ Dmax ≤ 35) | | | |
| 1gwc_00021** | wMBE | 6 | 8 | 14 | | timeout | 11.72 | 625.4 | 11.72 |
| | wMBE+ | 6 | 8 | 14 | 19.92 | 1511.9 | 11.92 | | |
| 2hnv_00025** | wMBE | 6 | 10 | 18 | | timeout | 11.52 | 1013.2 | 13.65 |
| | wMBE+ | 6 | 10 | 18 | 21.81 | 2085.6 | 16.18 | | |
| 2rf9_00007** | wMBE | 8 | 11 | 20 | | timeout | 12.02 | 399.6 | 14.73 |
| | wMBE+ | 8 | 11 | 20 | 515.99 | timeout | 13.41 | | |
| 2rf9_00013** | wMBE | 6 | 10 | 18 | | 1686.5 | 15.03 | 43.8 | 15.03 |
| | wMBE+ | 5 | 10 | 18 | 17.68 | 62.3 | 15.03 | | |
| 2rfe_00017** | wMBE | 5 | 9 | 16 | | timeout | 10.33 | 91.2 | 10.80 |
| | wMBE+ | 5 | 9 | 16 | 23.00 | timeout | 10.86 | | |
| 2rfe_00030** | wMBE | 6 | 9 | 16 | | timeout | 10.27 | 248.3 | 10.97 |
| | wMBE+ | 6 | 9 | 16 | 515.23 | timeout | 10.29 | | |

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
