# OpenReview forum: "AND/OR Branch-and-Bound for Computational Protein Design Optimizing K*"
_auai.org/UAI/2022/Conference — UAI 2022 Poster_

### Official Review · Reviewer_p6zu · 2022-04-07

**Q2(1) Originality/Novelty:** 3
**Q2(2) Significance/Impact:** 3
**Q2(3) Correctness/Technical Quality:** 3
**Q2(6) Clarity Of Writing:** 2
**Q6 Overall Score:** 7
**Q8 Confidence In Your Score:** 3

**Q1 Summary And Contributions:**

They apply the AND/OR Branch-and-Bound framework on a variation of Marginal MAP, optimizing K\* (protein design application). The distinction of K\*MAP is a specific quotient operation, requiring bounds for each subunit. They adapt a mini bucket elimination and AND/OR BnB (AOBB) algorithm, propose 2 encodings of the protein design problem into their K\*MAP setting, and experiment with using a weighted heuristic, infusing thresholds and show favorable results against existing solver BBK\*.

**Q2 Assessment Of The Paper:**

More detailed information regarding each of these aspects is given below:

**Q2(4) Quality Of Experiments (Optional):**

3: Good: The experimental evaluation is adequate, and the results convincingly support the main claims.

**Q2(5) Reproducibility:**

2: Fair: Key resources (e.g., proofs, code, data) are unavailable but key details (e.g., proof sketches, experimental setup) are sufficiently well-described for an expert to confidently reproduce the main results.

**Q3 Main Strengths:**

The paper tackles a protein design problem, which has recently gained quite some interest, by adapting a well known framework of AND/OR Branch and Bound. I believe this will be of interest to the UAI community. (**+Impact**) Results, comparing to an existing solver, appear very positive. A variety of optimisations have been considered and analysed in the experiments. (**+Quality of experiments**)

**Q4 Main Weakness:**

The problem description is not super clear for readers unfamiliar to protein design. Examples could help. The supplementary material alleviated some of this issue (Sect 2.1, 2.2, Fig 5, 6).

Comparison with state-of-the-art could be improved. A comparison is made to BBK* but the explanation of the algorithmic differences is limited.

**Q5 Detailed Comments To The Authors:**

## Questions


Q1) The paper mentions that the used benchmarks were obtained by ANONYMOUS. Will the used data set be publicly released for reproducability and algorithm improvement purposes? Are there no publicly standardized benchmarks for this problem yet? https://arxiv.org/pdf/2202.01079.pdf seems to talk about this topic, are there any specific arguments for the data set used in the experiments of this paper? For example, why not use the benchmarks in the $BBK^*$ paper?


Q2) In Algorithm 1 line 12 and 16 the inner power is $w_t$ and the outer is $1/w_t$. In Marinescu et al. [2014]. (Algorithm 3), and the definition of power-sum this seems to be the opposite, first $1/w_t$ and then $w_t$?


Q3) The $(wt)$ in $K^{*(wt)}$ and $Z^{(wt)}$ is confusing me. I assume it's not to the power of $(wt)$ and it is instead an 'identifier/index'. But is the $w$ related to $\mathbf{w} = \{w_1,\dots,w_T\}$? The $Z^{(wt)}$ is "the partition function of the **wild-type** sequence" but I did not quite understand the wild-type part, nor how $w$ is related here.


Q4) The experiments compare against $BBK^*$, "a state-of-the-art best-first search algorithm available in OSPREY 3.0.". How is $BBK^*$ different from $AOBB-K^*$? Is best-first vs depth-first the only difference?


Q5) Tabel 2: the cases were F2 is slower than F1 is due to the pre-t. The paper mentions this is "(ex. compiling heuristics)", can you elaborate on what computations may cause this increase for F2 here?

Q6) Section 6. paragraph AOBB-K* vs BBK* (solution quality). Theorem 5.1 states that AOBB-K* returns the optimal K* value. Is BBK* an approximate method then or why do they not find the same best values?

Q7) Table 1: Theorem 5.1 states AOBB-K* returns the optimal K* value. In Table 1 the best found K* was different for F1 than F2? If this is to due stopping the process after 1hr, what is the difference between AOBB-K* and any-AOBB-K*?

Q8) Table 3, the user parameter iB is different per benchmark instance. How was this value decided?

Q9) Table 5 is about comparing the w=0.001 vs w=1 parameter. To estimate the impact of this parameter, shouldn't both cases always use the same user parameter iB (cf. 2rfe\_00017* and 4wwi\_00019*) to compare their performance?

Q10) Table 6 same question as Q9 but with the underflow parameter.



## Textual remarks

Section 3. Objective paragraph, explicitly put $argmax_R$, $argmax_R K^*(R_1...R_n)$

Section 4 "made by partitioning the bucket functions into mini-buckets" -> "...into $T$ mini-buckets"

Section 4 The part about taking the product of their power-sums, leveraging Holder's Inequality is very confusingly written for those that do are not familiar with the Inequality. The equation written there is the definition used by Marinescu et al. 2014 to denote a 'power-sum', which they then later use in the Holder's Inequality. Since the power sums are seen in line 12 and 16, I suggest to drop the power-sum equation in the text to reduce confusion.

Section 5.2 first sentence, place parenthesis around citation Dechter [2019]?

Section 6, bottom line "AOBB-MMAPMarinescu et al." -> space between citation and word.

Section 6, "statistics in Table 1that" -> "..Table 1 that"

Section 6, "in Table 1show" -> "in Table 1 show"


**Supplementary material**

Sect 1: "is in different structural states...", ...?

Sect 6 "an depth-first" -> "a depth-first"

Sect 7.1 "who's function" -> "whose function"

Sect 7.1 At the of the line of $Z^{min}_B$ there is something wrong with the paranthesis and superscript 1.

Sect 10 is a duplicate of Sect 9?

------

# After Rebuttal

I thank the authors for their clarifications. I have also read the other reviews and the authors' replies, and will stick to my current evaluation.

It would be good if the authors could revise the paper to elaborate on the differences with BBK* (as they did in the rebuttal) and clarify the problem description a bit more based on the review feedback.

**Q7 Justification For Your Score:**

The authors address an interesting problem, by adapting an approach known and interesting to the UAI community. I am inclined to accept, but would like to see the authors elaborate on the differences with prior work ($BBK^*$), (in rebuttal and paper if accepted).

**Q9 Complying With Reviewing Instructions:**

1: Yes.

---

### Official Review · Reviewer_2TZ1 · 2022-04-08

**Q2(1) Originality/Novelty:** 3
**Q2(2) Significance/Impact:** 3
**Q2(3) Correctness/Technical Quality:** 3
**Q2(6) Clarity Of Writing:** 3
**Q6 Overall Score:** 7
**Q8 Confidence In Your Score:** 3

**Q1 Summary And Contributions:**

The paper presents a new graphical model inference problem: estimating the ratio of two marginal partition functions corresponding to partial settings of a subset of variables. This is applicable to protein design to estimate the ratio of energies of two conformations, which corresponds to the relative probability of being in the two conformations.

This highly technical paper is carefully situated in a deep line of work on sophisticated methods for marginal-map inference.

**Q2 Assessment Of The Paper:**

More detailed information regarding each of these aspects is given below:

**Q2(4) Quality Of Experiments (Optional):**

3: Good: The experimental evaluation is adequate, and the results convincingly support the main claims.

**Q2(5) Reproducibility:**

4: Excellent: Key resources (e.g., proofs, code, data) are available and key details (e.g., proof sketches, experimental setup) are comprehensively described for competent researchers to confidently and easily reproduce the main results.

**Q3 Main Strengths:**

I am not an expert on this particular line of algorithms for marginal-map inference and associated alternative problems, so it's difficult for me to evaluate the correctness of the algorithm details. I know that they are popular at UAI and there is a healthy community of research on it.

I do have considerable background on undirected graphical models and protein design however. I think that the problem formulation here is very well motivated.

Overall, I expect that the paper will be well received by the UAI community, since it elegantly extends methods developed for marginal-map to the paper's new problem definition. I also found the evaluation comprehensive and illuminating. I appreciate the exploration of the two problem formulations. The extensions to the new problem formulation are non-trivial and require deep familiarity with this area.

**Q4 Main Weakness:**

The paper could do a better job of situating their particular approach within the field of computational protein design. They tackle a very specific problem formulation. Computational protein design is a very broad field, that includes things such as AlphaFold. I'd use https://github.com/yangkky/Machine-learning-for-proteins for pointers to some reviews and update the introduction section.


**Q5 Detailed Comments To The Authors:**


It would be helpful to have further discussion of qualitatively-different approaches to estimating this ratio. What could be done, for example, with MCMC. I don't suggest including these in the experiments, since the experiments are focused on ablations within a particular line of approaches. However, it would be to understand why this particular family of approaches is the best.



**Q7 Justification For Your Score:**

The paper is part of a line of high-quality work that frequently appears at UAI. The paper is technically sound (with an extremely comprehensive supplementary material!). The problem definition is well motivated, interesting, and requires non-trivial adaption of existing algorithms.

**Q9 Complying With Reviewing Instructions:**

1: Yes.

---

### Official Review · Reviewer_6GTG · 2022-04-11

**Q2(1) Originality/Novelty:** 2
**Q2(2) Significance/Impact:** 2
**Q2(3) Correctness/Technical Quality:** 2
**Q2(6) Clarity Of Writing:** 1
**Q6 Overall Score:** 3
**Q8 Confidence In Your Score:** 2

**Q1 Summary And Contributions:**

The paper focuses on computational protein design, with the goal to optimize an (approximate) predicted affinity K* between proteins by varying (parts of) protein sequence. Specifically, the goal is to find the maximum a posteriori (MAP) estimation of K* (which is basically a Bayesian MLE). This is difficult since every protein seq. can exist in many different configurations with each their own K, such that the search space is very large. The authors present algorithms to find K*MAP efficiently.

**Q10 Ethical Concerns (Optional):**

None other than the potential plagiarism in the introduction.

**Q2 Assessment Of The Paper:**

More detailed information regarding each of these aspects is given below:

**Q2(5) Reproducibility:**

1: Poor: Key details (e.g., proof sketches, experimental setup) are incomplete/unclear, or key resources (e.g., proofs, code, data) are unavailable.

**Q3 Main Strengths:**

Like the authors mention, computational protein design is indeed a relevant topic with many potential applications across biology. But it is also a challenging task, and as far as I know, computational design cannot yet replace experimental testing of protein candidates. If algorithms to predict binding affinity can be improved to eventually replace time-consuming laboratory tests, this would indeed be an important contribution.

**Q4 Main Weakness:**

1)	The beginning of the introduction is almost word for word the same as of: https://ojs.aaai.org/index.php/AAAI/article/view/6584. This seems to be plagiarism. (Or self-plagiarism, which I cannot assess given anonymity – but as far as I know self-plagiarism remains plagiarism if it is not clear in the paper that this is a direct quote from another paper).

2)	As far as I can tell, the authors do not mention making available their code or the problems they evaluate their algorithms on (e.g. which proteins, which mutable residues, which rotamers considered, which constraints used in F2). This is not trivial to reproduce.

3)	The paper is quite difficult to read (at least for people who are not experts in this exact field); see #1 in Q5 for some concrete aspects that were difficult to grasp.

4)	The exact aim was unclear to me, as well as the relation to existing work. This makes it hard to assess impact. See comment #3 in the detailed comments for an explanation/suggestions.


**Q5 Detailed Comments To The Authors:**

1)	I found the paper quite difficult to understand. This is partially due to the structure, where many things are mentioned before they are explained. For example:
- The introduction immediately dives into MMAP and KMAP, without explaining what MAP is or why you would want to do this in this context. The larger context (CPD) is then only explained in section 2; MAP is explained nowhere at all.
-	The explanation of K in 2.2 depends on detailed knowledge of Bolzmann distributions/statistical thermodynamics, which might be a bit specialistic for the audience (e.g. I am not sure whether you can consider “partition functions” or “entropic contributions” here common knowledge).
It might be easier to immediately present eqn 1 as an approximation of K (with reference to the paper and perhaps a book on statistical thermodynamics, or an appendix which goes into the details).
-	After considering CPD and K*MAP in 2.1/2.2, section suddenly goes back to the general MMAP 2.3, explaining different types of graphical structures and how to construct a pseudo tree. But at this point I as a reader had no clue:
a) What the nodes in the pseudo tree (and other graphs) represent, and how this relates to the KMAP or even MMAP problem;
b)	Why we need the pseudo tree in the first place.

-	Section 3 goes back to KMAP and explains the two formulations, which essentially define how to compute K. But how is this a “graphical formulation”? – i.e. it is not clear how you get from a formula for K to a graphical structure (as the section title seems to suggest).

2)	The paper introduces KMAP as an analogue of MMAP. But I am not sure I truly understand the link between MMAP and KMAP:

-	In MMAP, we have some x that depends (probabilistically) on parameters theta and phi, such that we have a probability distribution: f( x | theta, phi), and we want to find the arg max_theta over f( x | theta ) marginalized over phi, with f( x | theta ) = Sum_phi f( x | theta, phi ).

-	In K\*MAP, we consider K\*, and we again have variables we maximize over: (1) the residues R, presented by the authors as analogous to theta, and (2) the configurations C, presented as analogues of phi.

-	However, the presented K\* seems to be only a function of R, and not of C: summing over the configurations C is part of the definition for K\*, such that we end up with a formula that only depends on R. So how is this analogous? And what exactly does this mean? Because conceptually, K* should indeed be a function of both R and C since different configurations will have different binding energies and thus different affinities to each other.

-	Furthermore, as far as I understand from the text, if we know the residues R, then K\* is perfectly defined as a single value, so K* = f(R), rather than following a probability distribution f( K* | R). Again, what is the analogy?

Could you explain this?

3)	It is not clear to me whether the main idea of the paper is to introduce K*MAP itself, or the algorithms to compute it. The introduction seems to suggest the former (We introduce… K*MAP), but the list of contributions and the end of section 2.2 suggests the latter (The goal…develop efficient algorithms for K*MAP).
a.	If this is indeed the first paper introducing K*MAP, the authors should probably corroborate their premise that K*MAP is better than the mentioned state of the art (GMEC), by comparing the two approaches (see also below).
b.	If K*MAP has been introduced previously, then these references should be cited
c.	Is the main goal to make algorithms that find better values for K*MAP, algorithms that do it faster, or both? That should be clear from the beginning of the paper.
d.	Compared to GMEC, how does K*MAP compare in terms of:
i.	Quality of the solution
ii.	Time to find a solution
iii.	Types of problems that are feasible to solve (e.g. problem size, and discussing how this relates to typical problems from the CPD field)
These are important to know to assess potential impact of the paper.


**Q7 Justification For Your Score:**

Mostly 1,3, and 4 in Q4. I did not check the algorithms themselves and the empirical evaluation (which is beyond my technical expertise), so I focused mostly on the rationale and general concept (sections 1-3).

**Q9 Complying With Reviewing Instructions:**

1: Yes.

---

### Official Review · Reviewer_XhZn · 2022-04-11

**Q2(1) Originality/Novelty:** 3
**Q2(2) Significance/Impact:** 2
**Q2(3) Correctness/Technical Quality:** 3
**Q2(6) Clarity Of Writing:** 3
**Q6 Overall Score:** 7
**Q8 Confidence In Your Score:** 2

**Q1 Summary And Contributions:**

This paper applies previous research on AND/OR B&B search for the MMAP problem to a practical problem related to computational protein design that involves optimizing a measure called K* that approximates binding affinity.

**Q2 Assessment Of The Paper:**

More detailed information regarding each of these aspects is given below:

**Q2(4) Quality Of Experiments (Optional):**

3: Good: The experimental evaluation is adequate, and the results convincingly support the main claims.

**Q2(5) Reproducibility:**

3: Good: Key resources (e.g., proofs, code, data) are available and key details (e.g., proofs, experimental setup) are sufficiently well-described for competent researchers to confidently reproduce the main results.

**Q3 Main Strengths:**

The paper applies a well-developed heuristic search approach to MMAP, a classic problem of probabilistic inference, to an interesting and challenging bioinformatics problem.

**Q4 Main Weakness:**

I cannot identify any major weakness. However, it is a difficult-to-read paper that provides not much in the way of high-level insights, and mostly (and perhaps necessarily) it describes complex details of the problem and search algorithm.

**Q5 Detailed Comments To The Authors:**

More than 40 pages of supplementary material — that seems like a lot!

Although I understand how the problem can be solved by an AND/OR graph search algorithm, I cannot quite understand why it cannot also be solved more simply by an OR graph search algorithm, either A* or some form of B&B. (I guess this is a question about the AND/OR graph search framework for solving MMAP, and not a question about this specific application.) Since a solution of the problem is simply an instantiation of the MAP variables, can't the problem be formulated as the search for a path in an OR graph?

**Q7 Justification For Your Score:**

It’s very nice to see this line of research applied to challenging practical problem. However, I do not have a background in bioinformatics, and was not quite able to follow the details of this particular application based on the description in the paper, and so I am not qualified to evaluate the potential impact of this work on this application.

**Q9 Complying With Reviewing Instructions:**

1: Yes.

---

### Decision · Program_Chairs · 2022-05-15

**Decision:**

Accept (Poster)

**Comment:**

Meta Review: Most reviewers appreciate the contributions of the paper, albeit different parts.  The presentation is still an issue: since the paper touches many topics, none all of the reviewers were able to appreciate the parts of the paper that went outside their scope.  For example, reviewers with a background on inference had a harder time with the biology, and the reviewers with a background in biology had a harder time with the inference.  Please take into consideration the feedback from the reviewers in order to improve the readability of the paper for these audiences.

There is general consensus also that the authors should take allegations of plagiarism more seriously, even if there were no bad intentions.
Even if this practice (copying basic sentences) is common in this line of work, the authors should remember that other fields (and reviewers) will take such practices much more seriously.  At the least, the authors should either (1) revise the opening of the introduction, so that it is not a direct copy, or (2) cite the prior paper.